# On the Detection of Fracture within Vibrating Beams Traversed by a Moving Force

Georgios I. Dadoulis  and George D. Manolis *

Laboratory for Experimental Mechanics, School of Civil Engineering, Aristotle University of Thessaloniki, GR-54124 Thessaloniki, Greece; dadoulis@civil.auth.gr
* Correspondence: gdm@civil.auth.gr; Tel.: +30-2310-995663

**Abstract:** In this work, we examine the influence of a crack in the span of a beam as it is being traversed by a point force with constant velocity. This problem presents two types of discontinuities: one spatial, where the crack is modelled as a discontinuity in the slope of the deflection curve of the beam, and a temporal one, with the former derived as the point force moves forward in time. The aim is to interpret time signals registered at a given node on the beam, either during the forced vibration or the free vibration regimes, by using the Gabor transform of the transient beam response so as to observe a pattern that alludes to the location of the discontinuity. Three analytical methods are examined, namely eigenvalue extraction, Laplace transformation and the transform matrix technique. A numerical example is presented using the Laplace transformation, where it is possible to detect the location of damage during the traverse of a point force across the bridge span. Validation studies of the methodology presented here can be conducted in the future, either through field measurements or through experimental setups, which constitutes an important step in realizing applications in structural health monitoring of civil engineering infrastructure.

**Keywords:** discontinuities; damage; cracks; vibrations; beams; eigenvalues; moving force; Laplace transform; time-frequency spectra; Gabor transform

## 1. Introduction

The presence of cracks in the web of steel and reinforced concrete beams serving as girders, bridge decks, pylons, etc. heavily influences their vibratory response, and this has been well documented in the past [1]. This is an important problem in fracture mechanics [2], as cracks tend to propagate rather rapidly in the presence of time-dependent loads, weaken the beam and eventually cause local failure. Furthermore, there are three basic crack modes to contend with [3], namely Mode I (an open/close crack), Mode II (in-plane sliding of the crack surfaces) and Mode III (out-of-plane sliding of the crack surfaces). For girders sustaining vertical loads, Mode I is prevalent and an engineering-type approach to modelling this crack is by insertion of springs in the locality of the crack [4]. The type of spring used depends on the specific discontinuity expected to develop, as will be discussed in Section 2 which follows. The most common way for modelling damage is placing a vertically Mode I crack in a beam's web, and to further assume a discontinuity in the slope of the beam at that location, i.e., in the first derivative of the transverse displacement function [5].

It is well understood that the presence of a crack in a structural member induces an extra flexibility which affects its dynamic response. Specifically, new harmonics are introduced in the response spectrum, especially if the crack opens and closes during vibrations [1]. Over the last few decades, literally hundreds of papers have appeared on this subject, most of them from the field of mechanical engineering since the vibration of shafts, turbine blades, etc. are important topics in power generation and in propulsion. Among early work on measurements conducted on beams, we mention [4], who examined

a cantilevered beam with a transverse crack extending along its width. By measuring the response at two different points on the beam vibrating in one of its natural modes, in conjunction with an analytical solution, it was possible to produce a good estimate of the crack location.

There has also been theoretical work based on cracked beam theory for the prediction of changes in the transverse vibrations of beams with a breathing (i.e., open-close) crack [5]. The analysis traces eigenfrequency changes due to a breathing edge-crack, which are shown to depend on the bi-linear character of the dynamic system. By breaking up the problem into associated linear problems solved over their respective domains, the two solutions are matched through the initial conditions. It was determined that changes in vibration frequencies for a fatigue-breathing crack are smaller than the ones caused by open cracks. This type of work has been continued to cover arbitrary types of discontinuities at arbitrary locations in Bernoulli–Euler vibrating beams [6]. The discontinuities induced by various cracks are modelled by Heaviside functions so as to express the modal displacement of the entire beam by a single function. This general modal displacement function is then solved by using the Laplace transformation. This general solution covers four types of modal shapes induced by the four basic types of discontinuities. Numerical results for a cantilevered beam excited by an actuator suggest that the variation of a driving-point anti-resonance frequency can be used to determine the location and size of crack, a technique that has obvious structural health monitoring (SHM) applications [7].

When modelling discontinuities in the domain of the definition of differential equations, as is the case with Bernoulli–Euler fourth-order ordinary differential equation for beam bending, it is necessary to introduce generalized functions, among which is the well-known Dirac delta function [8]. Generalized functions, also known as Schwartz distributions, make it possible to differentiate functions whose derivatives do not exist in the classical sense. More specifically, any locally integrable function has a distributional derivative. In our case, we are interested in modelling jump discontinuities on displacements and rotations, so that the Dirac delta function and its first distributional derivative appear in the new force terms on the right-hand side of the Bernoulli–Euler equation. Here, we further develop the use of generalized functions for discontinuities [9] in conjunction with the Laplace transform to demonstrate that it is possible to clearly identify the location of a crack in a beam from the Fourier transform of its transient response.

Moving loads across bridges and other structures [10,11] presents an important opportunity for detecting deterioration over time, either by the loads themselves or because of environmental conditions in general. In terms of the use of various analytical methodologies for determining and processing dynamic signals for SHM purposes, we mention the short-term Fourier transform (STFT), which in other fields of application is known as the Gabor transform [12]. Another promising technique for the processing of vibration data for damage identification is the wavelet transform [13]. Finally, use of the Hilbert transform as a means for analyzing the acceleration signals recorded over large time intervals in a single span ballasted railway bridge for tracing changes in the eigenfrequencies and damping ratios was reported in [14].

The possibility of detecting damage in bridges due to a reduction in stiffness was explored in [15], based on both 2D and 3D numerical simulations of vehicle—bridge interaction. To that purpose, the STFT was used to examine the energy band variation of the vehicle's acceleration time history, which was subsequently found to strongly correlate with damage parameters. It was also found that the vehicle's initial entering conditions were critical in obtaining the correct vehicle response. In terms of use of data recovered from vehicles travelling over R/C bridges, the relaxation of prestressing by modelling the bridge as a continuous beam with eccentric prestress was examined in [16]. The vehicle itself was represented as a four degree-of-freedom system, while the model developed was validated against numerical simulations using finite elements and against results appearing in the literature. Following parametric studies, it was shown that prestress influences the

maximum vertical acceleration of vehicles, a fact that can be used as an index for detecting the loss of prestress.

When it comes to conducting field measurements, one possibility is to use the moving vehicle as a message carrier for estimating the dynamic properties of the bridge [17]. Along these lines, the short time frequency domain decomposition (STFDD) has been used to estimate bridge mode shapes from the dynamic response of a running vehicle, where the bridge is segmented and measurements are performed using two instrumented axles [18]. It is noted here that the road profile may excite the vehicle, making detection of the bridge modes difficult. A recent application on the use of a single axle, two-wheel test vehicle to detect damage in a simply supported plate-type bridge at the two longitudinal ends, but free along the lateral sides, was reported in [19]. The test vehicle was represented as a two degree-of-freedom system to capture motions in both longitudinal and lateral directions, taking into account the separation of the vehicle's eigenfrequencies from those of the deck. Damage localization can also be detected from mode shapes extracted from the moving vehicle response in beam structures without any reference data, where the first-order mode shape with high spatial resolution in the damaged state is extracted from the response measured on a moving vehicle via the Hilbert transform [20]. Finally, an additional issue regarding damage detection in beams from their dynamic response due to the passage of a moving force is the decomposition of the acceleration response into "static" and "dynamic" components, followed by an additional "damage" component which corresponds to a localized loss in stiffness [21]. Thus, these three components combine to establish how a damage singularity will appear in the total response.

## 2. Discontinuities in Bernoulli-Euler Beams

We identify four possible discontinuities in a beam under flexure:

1. Displacement discontinuity at station $L_i$: $w\left(L_i^+, t\right) - w\left(L_i^-, t\right) = \Delta w(L_i, t) \neq 0$
2. Slope discontinuity at station $L_i$: $w'\left(L_i^+, t\right) - w'\left(L_i^-, t\right) = \Delta w'(L_i, t) \neq 0$
3. Bending moment discontinuity at station $L_i$: $w''\left(L_i^+, t\right) - w''\left(L_i^-, t\right) = \Delta w''(L_i, t) \neq 0$
4. Shear force discontinuity at station $L_i$: $w'''\left(L_i^+, t\right) - w'''\left(L_i^-, t\right) = \Delta w'''(L_i, t) \neq 0$

In the above, $w(x, t)$ is the time-dependent transverse displacement of the beam, while primes ($'$) indicate spatial derivatives ($\frac{d}{dx}$) and symbol $\Delta$ denotes the discontinuity (jump). Table 1 gives the schematics for these four cases and identifies the type of equivalent spring $K$ that could be used to model each particular discontinuity. From a mathematical viewpoint, there is a cascading effect in this hierarchy in the sense that a discontinuity in the slope of a function necessitates discontinuities in all subsequent higher derivatives.

**Table 1.** List of beam discontinuities.

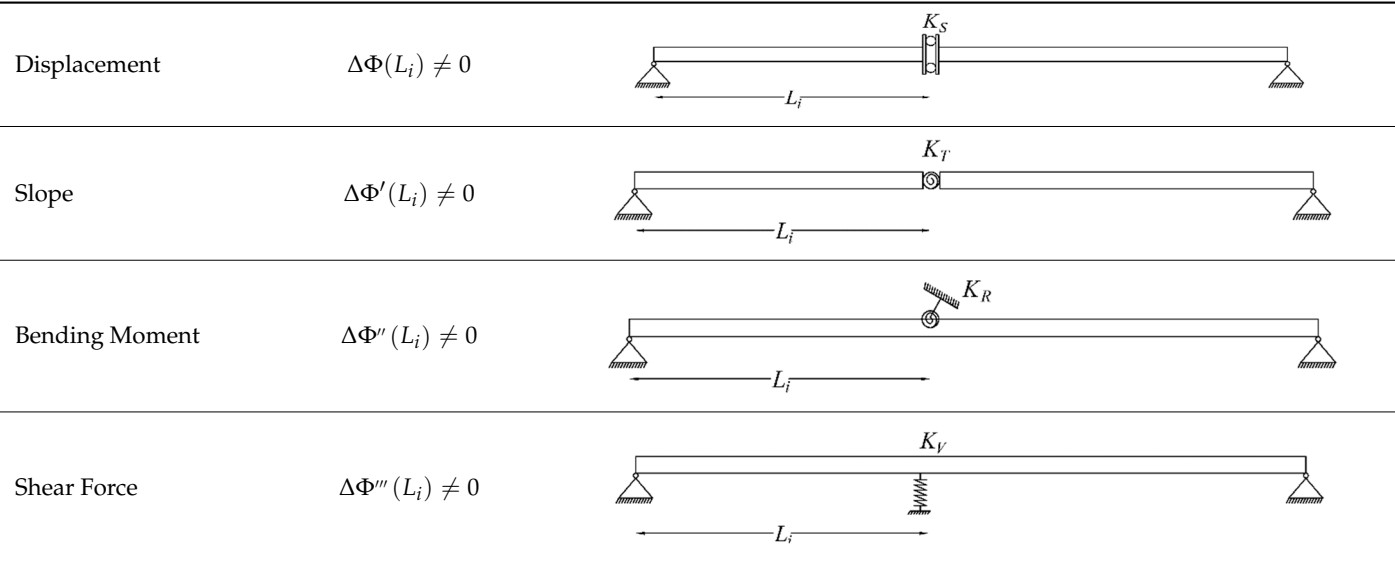

| | | |
|---|---|---|
| Displacement | $\Delta\Phi(L_i) \neq 0$ | $K_S$ ... $L_i$ |
| Slope | $\Delta\Phi'(L_i) \neq 0$ | $K_T$ ... $L_i$ |
| Bending Moment | $\Delta\Phi''(L_i) \neq 0$ | $K_R$ ... $L_i$ |
| Shear Force | $\Delta\Phi'''(L_i) \neq 0$ | $K_V$ ... $L_i$ |

## 3. Methods of Analysis for Beam Discontinuities

Three methods of analysis will be presented for the type 2 discontinuity at an interior point ($0 < x = L_i < L$) of a beam segment, namely, in the slope of the displacement function, as in Figure 1. This type is the most standard type of damage that leads to failure for vertical loads in the plane of the beam [3].

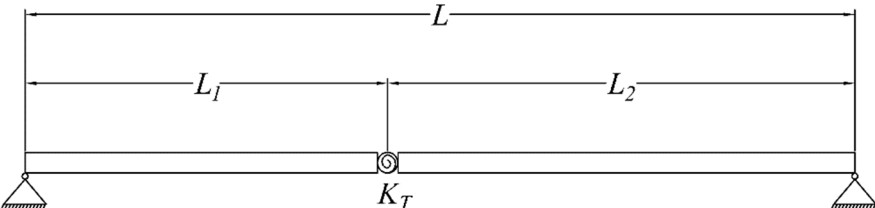

**Figure 1.** Simply supported beam segment with a slope discontinuity at station $L_1$.

At first, the following assumptions need to be made if the slope discontinuity is to correspond to a vertical crack in the beam's web:

1. The crack caused by the discontinuity modifies the stiffness of the beam locally, while the beam's mass remains unchanged.
2. The crack remains open, i.e., there is no contact between its two faces. This happens if the crack develops in the tension zone of the beam's web and the loads are of static nature. If we have dynamic loads, it is possible that the two crack faces come in contact during vibration, and the phenomenon becomes non-linear and beyond the scope of this work.
3. Only the local bending stiffness $EI$ of the beam is affected by the crack.
4. A numerical value for an equivalent spring $K_T$ at $x = L_i$ must be a priori estimated.

For example, an equivalent static spring value for $K_T$ can be computed for a beam with an orthogonal cross-section $b \cdot h$ under a bending moment $M(x)$ and for a vertical crack length $\alpha$. Use of Castigliano's theorem gives the crack movement $u = \frac{\partial U}{\partial P}$, where $U$ is the potential energy that develops because of load $P$, which in turn corresponds to the bending moment $M$. Therefore, $U = \int_0^a (\partial U/\partial\alpha)da = \int_0^a J(a)da$, where $J(a)$ is the energy density function of $U$. Combining these relations gives the crack movement as $u = \partial\left(\int_0^a J(a)da\right)/\partial M$. Finally, by defining the flexibility coefficient as $F = \frac{\partial u}{\partial M}$ yields $F = \frac{\partial^2 \left(\int_0^a J(a)da\right)}{\partial M^2}$. The energy density function $J(a)$ assumes different values for each beam

cross-section [2]. For an orthogonal cross-section, we compute $F = 5.346 \frac{h}{EI} f\left(\frac{a}{h}\right)$, where $f\left(\frac{a}{h}\right) = 1.8624 \left(\frac{a}{h}\right)^2 - 3.95 \left(\frac{a}{h}\right)^3 + 16.375 \left(\frac{a}{h}\right)^4 - 37.226 \left(\frac{a}{h}\right)^5 + 76.81 \left(\frac{a}{h}\right)^6 - 126.9 \left(\frac{a}{h}\right)^7 + 172 \left(\frac{a}{h}\right)^8 - 143.97 \left(\frac{a}{h}\right)^9 + 66.56 \left(\frac{a}{h}\right)^{10}$. Finally, the static spring stiffness is recovered as the inverse of flexibility, i.e., $K_T = \frac{1}{F}$.

### 3.1. Eigenvalue Extraction

Eigenvalue extraction is a standard method in structural dynamics [10]. For the simply supported beam of Figure 1 comprising two segments with total length $L = L_1 + L_2$, the *ith* eigenfunction $\Phi_i(x)$, $i = 1, 2, 3, \ldots$ can be broken into two parts corresponding to the two beam segments (left and right) as follows:

$$\Phi_{i1}(x) = a_1 \sin(k_i x) + a_2 \cos(k_i x) + a_3 \sinh(k_i x) + a_4 \cosh(k_i x), \quad 0 \leq x \leq L_1 \quad (1)$$

$$\Phi_{i2}(x) = b_1 \sin(k_i x) + b_2 \cos(k_i x) + b_3 \sinh(k_i x) + b_4 \cosh(k_i x), \quad 0 \leq x \leq L_2 \quad (2)$$

Eight boundary conditions are now required, starting with the two simply supported ends and adding compatibility at the common node (continuity in the displacement, the moment diagram, the shear diagram and the discontinuity in the slope):

$$\Phi_{i1}(L_1) = \Phi_{i2}(0), \ \Phi_{i1}''(L_1) = \Phi_{i2}''(0) \ , \ \Phi_{i1}'''(L_1) = \Phi_{i2}'''(0), \ \Phi_{i2}'(L_1) - \Phi_{i1}'(0) = \left(\frac{EI}{K_T}\right)\Phi_{i1}''(L_1) \quad (3)$$

From these conditions, an $8 \times 8$ homogeneous matrix system derives, whose determinant is set equal to zero, thus yielding wave numbers $k_i$, $i = 1, 2, 3, \ldots$ from which the eigenfrequencies $\omega_i$ are readily computed.

### 3.2. The Laplace Transform

This is a more general method [8] and requires a formal definition of the Heaviside function $H$ as a mathematical way for representing the discontinuity:

$$f(x) = f_1(x) + (f_2(x) - f_1(x))H(x - L_1) = f_1(x) + \Delta f(x) H(x - L_1) \quad (4)$$

As before, symbol $\Delta$ indicates a jump at location $x$ in function $f$. Two useful properties of the Heaviside function are $H'(x - L_1) = \delta(x - L_1)$ and $f(x) \delta(x - L_1) = f(L_1) \delta(x - L_1)$.

For the first and second parts of the beam in Figure 1, we have their corresponding governing equations of motion in the absence of an external force as follows:

$$\rho A \ddot{w}_1(x, t) + EI \, w_1''''(x, t) = 0 \quad (5)$$

$$\rho A \ddot{w}_2(x, t) + EI \, w_2''''(x, t) = 0 \quad (6)$$

Assuming that is a slope discontinuity at station $L_1$, the displacement function can be written as

$$w(x, t) = w_1(x, t) + \Delta w(x, t) H(x - L_1), \ \Delta w(x, t) = w_2(x, t) - w_1(x, t) \quad (7)$$

The first four spatial derivatives of the above displacement function are

$$w'(x, t) = w_1'(x, t) + \Delta w'(x, t) H(x - L_1) + \Delta w(L_1, t) \delta(x - L_1) \quad (8)$$

$$\begin{aligned} w''(x, t) = w_1''(x, t) + \Delta w''(x, t) H(x - L_1) \\ + \Delta w'(L_1, t) \delta(x - L_1) + \Delta w(L_1, t) \delta'(x - L_1) \end{aligned} \quad (9)$$

$$\begin{aligned} w'''(x, t) = w_1'''(x, t) + \Delta w'''(x, t) H(x - L_1) \\ + \Delta w''(L_1, t) \delta(x - L_1) + \Delta w'(L_1, t) \delta'(x - L_1) \\ + \Delta w(L_1, t) \delta''(x - L_1) \end{aligned} \quad (10)$$

$$w''''(x,t) = w_1''''(x,t) + \Delta w''''(x,t)\, H(x-L_1) + \Delta w'''(L_1,t)\delta(x-L_1)$$
$$+\Delta w''(L_1,t)\delta'(x-L_1) + \Delta w'(L_1,t)\delta''(x-L_1) \tag{11}$$
$$+\Delta w(L_1,t)\delta'''(x-L_1)$$

By combining Equations (5) and (6) where the subscripts $1, 2$ refer to the two beam segments, we have

$$w_1''''(x,t) + \frac{\rho A}{EI}\ddot{w}_1(x,t) = \left(w_2''''(x,t) + \frac{\rho A}{EI}\ddot{w}_2(x,t) - \left(w_1''''(x,t) + \frac{\rho A}{EI}\ddot{w}_1(x,t)\right)\right)H(x-L_1) \tag{12}$$

Rearranging the above equation yields:

$$w_1''''(x,t) + H(x-L_1)\,\Delta w''''(x,t) = -\left(\frac{\rho A}{EI}\right)\left(\ddot{w}_1(x,t) + \Delta\ddot{w}(x,t)H(x-L_1)\right) = 0 \tag{13}$$

Since $\ddot{w} = \ddot{w}_1(x,t) + \Delta\ddot{w}(x,t)H(x-L_1)$, then

$$w_1''''(x,t) + \Delta w''''(x,t)H(x-L_1) = -(\rho A/EI)\ddot{w}(x,t) = 0 \tag{14}$$

Inserting Equation (11) in Equation (14) yields:

$$w''''(x,t) + \left(\frac{\rho A}{EI}\right)\ddot{w}(x,t) = \Delta w'''(L_1,t)\delta(x-L_1) + \Delta w''(L_1,t)\delta'(x-L_1)$$
$$+\Delta w'(L_1,t)\delta''(x-L_1) + \Delta w(L_1,t)\delta'''(x-L_1) \tag{15}$$

It is now possible to use the separation of variables for the displacement function in terms of the generalized coordinates $q(t)$ and the eigenfunctions $\Phi(x)$ in Equation (15) as $w(x,t) = \Phi(x)q(t)$ to recover

$$\ddot{q}(t) = -\omega^2 q(t) \tag{16}$$

$$\Phi''''(x) - k^4\Phi(x) = \Delta\Phi'''(L_1)\delta(x-L_1) + \Delta\Phi''(L_1)\delta'(x-L_1) +$$
$$\Delta\Phi'(L_1)\delta''(x-L_1) + \Delta\Phi(L_1)\delta'''(x-L_1) \tag{17}$$

The wavenumber appearing in the second equation for the eigenfunction is $k^4 = \rho A\omega^2/EI$. By applying the Laplace transform to Equation (17), with respect to the spatial variable $x$, where $s$ is the Laplace transform parameter, solving the resulting algebraic equation for the transformed eigenfunction $\Phi(s)$ yields

$$\Phi(s) = \frac{s^3}{s^4-k^4}\Phi(0) + \frac{s^2}{s^4-k^4}\Phi'(0) + \frac{s}{s^4-k^4}\Phi''(0) + \frac{1}{s^4-k^4}\Phi'''(0) + \frac{s^3 e^{-sL_1}}{s^4-k^4}\Delta\Phi(L_1) + \frac{s^2 e^{-sL_1}}{s^4-k^4}\Delta\Phi'(L_1) +$$
$$\frac{s\,e^{-sL_1}}{s^4-k^4}\Delta\Phi''(L_1) + \frac{e^{-sL_1}}{s^4-k^4}\Delta\Phi'''(L_1) \tag{18}$$

The above form of the solution allows for a closed-form inverse Laplace transformation [22] valid across the entire beam length $0 \le x \le L$ as follows:

$$\Phi(x) = \Phi(0)\,S_0(x) + \Phi'(0)S_1(x) + \Phi''(0)S_2(x) + \Phi'''(0)S_3(x) +$$
$$(\Delta\Phi(L_1)S_0(x-L_1) + \Delta\Phi'(L_1)S_1(x-L_1) + \Delta\Phi''(L_1)S_2(x-L_1) + \Delta\Phi'''(L_1)\,S_3(x-L_1))H(x-L_1) \tag{19}$$

where $S_0(x) = \frac{1}{2}(\cosh kx + \cos kx)$, $S_1(x) = \frac{1}{2k}(\sinh kx + \sin kx)$, $S_2(x) = \frac{1}{2k^2}(\cosh kx - \cos kx)$, and $S_3(x) = \frac{1}{2k^3}(\sinh kx - \sin kx)$.

Equation (19) is a general expression for an eigenfunction in the presence of a discontinuity at station $x = L_1$. Additional discontinuities at stations $x = L_i$, $i = 1, N$ can be superimposed to produce a more general eigenfunction form as follows:

$$\Phi(x) = \Phi(0)S_0(x) + \Phi'(0)S_1(x) + \Phi''(0)S_2(x) + \Phi'''(0)S_3(x) +$$
$$\sum_{i=1}^{N}(\Delta\Phi(L_i)S_0(x-L_i) + \Delta\Phi'(L_i)S_1(x-L_i) + \Delta\Phi''(L_i)S_2(x-L_i) + \Delta\Phi'''(L_i)\,S_3(x-L_i))H(x-L_i) \tag{20}$$

In sum, at station $x = L_i$ one may have discontinuity in the displacement, the slope, the bending moment and the shear force, as was shown in Table 1.

Focusing on the crack as modelled by a slope discontinuity at $x = L_1$, we can write that

$$\Phi'\left(L_1^+\right) - \Phi'\left(L_1^-\right) = \Delta\Phi'(L_1) = M\left(L_1^+\right)/K_T = M\left(L_1^-\right)/K_T = \frac{EI\Phi''\left(L_1^-\right)}{K_T} = \frac{EI\Phi''\left(L_1^+\right)}{K_T} \tag{21}$$

Listed below are the spatial derivatives of the eigenfunctions containing a discontinuity and are necessary in representing the cracks in the beams:

$$\Phi(x) = \Phi'(0)S_1(x) + \Phi'''(0)S_3(x) + \Delta\Phi'(L_1)S_1(x - L_1)\,H(x - L_1)$$
$$\Phi'(x) = \Phi'(0)\,S_1'(x) + \Phi'''(0)\,S_3'(x) + \left(\Delta\Phi'(L_1)\,S_1'(x - L_1)\right)H(x - L_1)$$
$$\Phi''(x) = \Phi'(0)S_1''(x) + \Phi'''(0)\,S_3''(x) + \left(\Delta\Phi'(L_1)\ S_1''(x - L_1)\right)H(x - L_1) + \Delta\Phi'(L_1)\,\delta(x - L_1)$$
$$\Phi'''(x) = \Phi'(0)S_1'''(x) + \Phi'''(0)\,S_3'''(x) + \left(\Delta\Phi'(L_1)\ S_1'''(x - L_1)\right)H(x - L_1) + \Delta\Phi'(L_1)\,\delta'(x - L_1)$$
$$\Phi''''(x) = \Phi'(0)S_1''''(x) + \Phi'''(0)\,S_3''''(x) + \left(\Delta\Phi'(L_1)\,S_1''''(x - L_1)\right)H(x - L_1) + \Delta\Phi'(L_1)\,\delta''(x - L_1)$$

By manipulating the above equations and using the boundary conditions for simply-supported beam ends $\Phi(L) = 0$ and $\Phi''(L) = 0$ we can express the discontinuity at $x = L_1$ as a function of the end conditions:

$$\Delta\Phi'(L_1) = (EI/K_T)\left(\Phi'(0)S_1''(L_1) + \Phi'''(0)\,S_3''(L_1)\right) \tag{22}$$

Then, the eigenvalue problem formulation from which the wavenumbers $k_1, k_2, k_3, \ldots$ the beam with the slope discontinuity can be computed comes from setting the determinant of the 3x3 system matrix equal to zero:

$$\begin{bmatrix} S_1(L) & S_3(L) & S_1(L - L_1) \\ S_1'''(L) & S_3''(L) & S_1''(L - L_1) \\ S_1''(L_1) & S_3''(L_1) & -K_T/EI \end{bmatrix} \begin{bmatrix} \Phi'(0) \\ \Phi'''(0) \\ \Delta\Phi'(L_1) \end{bmatrix} = \begin{bmatrix} 0 \\ 0 \\ 0 \end{bmatrix} \tag{23}$$

### 3.3. The Transfer Matrix Method

Transfer matrices [23] are efficient when discontinuities appear sequentially along the length of a beam. We will examine two such discontinuities in the beam shown in Figure 2.

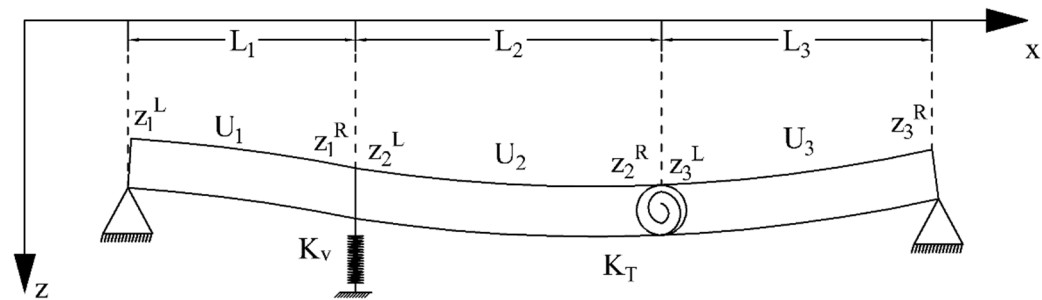

**Figure 2.** Simply supported beam with discontinuities at stations $L_1$ and $L_2$.

In order to transmit information from one station $z_i^L$ to the next $z_i^R$ within a beam segment $i$ moving from left to right, a transfer matrix $[T_i]$ is necessary. Next, transmission of information from segment $i$ to the adjoining segment $i + 1$ is accomplished by a new matrix $[C_i]$. We therefore have the sequence

$$\left\{z_1^R\right\} = [T_1]\left\{z_1^L\right\}, \quad \left\{z_2^L\right\} = [C_1]\left\{z_1^R\right\}, \quad \left\{z_2^R\right\} = [T_2]\left\{z_2^L\right\}, \quad \left\{z_3^L\right\} = [C_2]\left\{z_2^R\right\}, \quad \left\{z_3^R\right\} = [T_3]\left\{z_3^L\right\} \tag{24}$$

The above can be condensed to read as follows:

$$\left\{z_3^R\right\} = [A(k)]\left\{z_1^L\right\}, \quad [A(k)] = [T_3][C_2][T_2][C_1][T_1] \tag{25}$$

Thus, by setting the $\det[A(k)] = 0$ we can recover a sequence of wave numbers $k_1, k_2, k_3, \ldots$ and compute the corresponding eigenfrequencies. Of course, boundary conditions must be imposed, and, for homogeneous ones, the corresponding rows and columns in $A(k)$ must be deleted. For the particular example considered here, this reduces the size of the matrix from $4 \times 4$ to $2 \times 2$. No matter how many discontinuities are interposed, which affect the elements of matrices $C_i$, the system matrix $A(k)$ drops in size to a $2 \times 2$ for a single beam.

For the single slope discontinuity discussed previously, we have for $z_i^R = T_i z_i^L$, and thus recover the following form:

$$z_i^R = \begin{bmatrix} \Phi_i(L_i) \\ \Phi_i'(L_i) \\ \Phi_i''(L_i) \\ \Phi_i'''(L_i) \end{bmatrix} = \begin{bmatrix} S_0(L_i) & S_1(L_i) & S_2(L_i) & S_3(L_i) \\ S_0'(L_i) & S_1'(L_i) & S_2'(L_i) & S_3'(L_i) \\ S_0''(L_i) & S_1''(L_i) & S_2''(L_i) & S_3''(L_i) \\ S_0'''(L_i) & S_1'''(L_i) & S_2'''(L_i) & S_3'''(L_i) \end{bmatrix} \begin{bmatrix} \Phi_i(0) \\ \Phi_i'(0) \\ \Phi_i''(0) \\ \Phi_i'''(0) \end{bmatrix} = z_i^L \quad (26)$$

The second transfer matrix across beam segments $[C_1]$ for the case of a flexible support is

$$[C_1] = \begin{bmatrix} 1 & 0 & 0 & 0 \\ 0 & 1 & 0 & 0 \\ 0 & 0 & 1 & 0 \\ -K_V/EI & 0 & 0 & 1 \end{bmatrix} \quad (27)$$

In addition, the second transfer matrix $[C_2]$ for an internal joint (in our case, the discontinuity) is

$$[C_2] = \begin{bmatrix} 1 & 0 & 0 & 0 \\ 0 & 1 & EI/K_T & 0 \\ 0 & 0 & 1 & 0 \\ 0 & 0 & 0 & 1 \end{bmatrix} \quad (28)$$

## 4. Methods of Analysis for Beam Discontinuities

A simply supported beam of length $L = L_1 + L_2 = 5.0$ m consisting of a single steel section with an orthogonal cross-section of dimensions $b = 20$ cm, $h = 5$ cm representing a bridge girder is shown in Figure 3. A crack forms at station $L_1 = 2.0\ m$ measured from the left support. Following the Laplace transform method of analysis, Table 2 gives the first four eigenfrequencies $f_i$ (Hz) of the beam (i) without a crack, (ii) with a small vertical crack 10 mm deep and equidistant from the top and bottom surfaces of the beam and (iii) a larger crack that is 25 mm deep. The eigenfrequencies are computed by recourse to Equation (23).

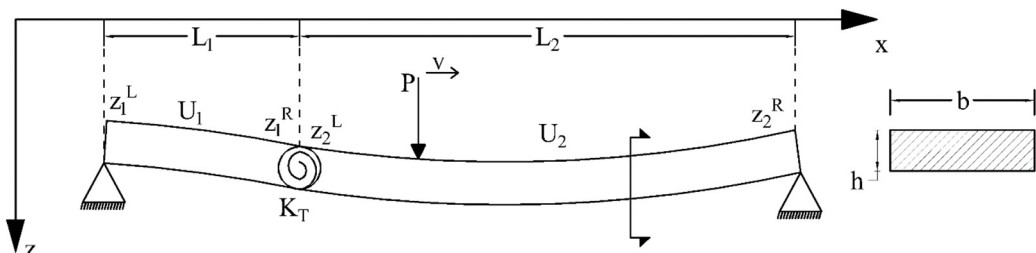

**Figure 3.** Simply supported beam with crack at station, $L_1$ modelled as a slope discontinuity by inserting a rotational spring $K_T$.

**Table 2.** First four eigenfrequencies $f_i$ of the simply supported beam with and without a crack.

| Crack Length $\alpha$ [mm] | Spring $K_T$ [kNm/rad] | $f_1$ [Hz] | $f_2$ [Hz] | $f_3$ [Hz] | $f_4$ [Hz] |
|---|---|---|---|---|---|
| 0.0 | $\infty$ | 4.58 | 18.31 | 41.20 | 73.24 |
| 10.0 | 12,814.9 | 4.55 | 18.27 | 41.11 | 72.82 |
| 25.0 | 1573.6 | 4.37 | 18.01 | 40.52 | 70.38 |

Next, Figure 4 plots the corresponding four eigenfunctions $\Phi_i(x)$, as well as their first and second derivatives, following a normalization in the form of $\rho A \int_0^L \Phi_i^2(x)dx = 1$. This results in the following values for the derivatives of the eigenfunctions at the origin: $\Phi_1'(0) = 1.419$, $\Phi_2'(0) = 2.701$, $\Phi_3'(0) = 4.453$, $\Phi_4'(0) = 5.212$. The formulas for the normalized eigenfunctions are now summarized below

$$\Phi_i(x) = \Phi_i'(0)\left(S_1(k_ix) + \frac{\Phi_i'''(0)}{\Phi_i'(0)}S_3(k_ix) + \frac{\Delta\Phi_i'(L_1)}{\Phi_i'(0)}S_1(k_i(x-L_1))\,H(x-L_1)\right) \quad (29)$$

where,

$$\frac{\Phi_i'''(0)}{\Phi_i'(0)} = -\frac{S_1''(L) + \frac{EI}{K_T}S_1''(L_1)S_1''(L-L_1)}{S_3''(L) + \frac{EI}{K_T}S_3''(L_1)S_1''(L-L_1)}, \quad \frac{\Delta\Phi_i'(L_1)}{\Phi_i'(0)} = \frac{EI}{K_T}\left(S_1''(L) + \frac{\Phi_i'''(0)}{\Phi_i'(0)}S_3''(L_1)\right)$$

$$(30)$$

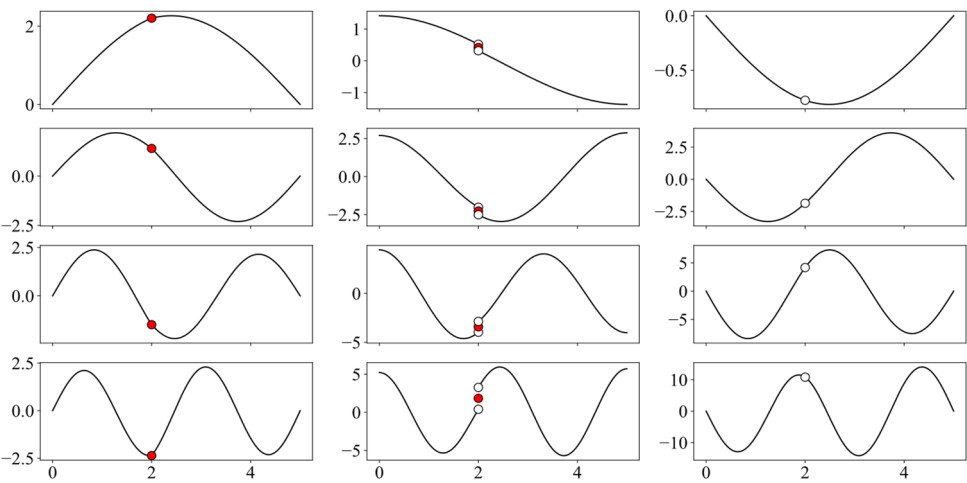

**Figure 4.** Normalized eigenfunctions for a simply supported beam of length $L = 5$ m, cross-section dimensions $b = 0.20$ m, $h = 0.05$ m and a spring value $K_T = 1573.6$ kNm/rad representing a vertical crack of length $\alpha = 25$ mm at station $L_1 = 2.00$ m. The first column is for eigenfunctions $\Phi_i(x)$, the second column is for the first derivatives $\Phi_i'(x)$ and the third column is for the second derivatives $\Phi_i''(x)$.

We note in Figure 4 that the slope discontinuity in the eigenfunction's first derivative at $L_1$ (red circle) results in an indefinite value for the second derivative, as indicated by a white circle.

Following the eigenvalue analysis, we now examine the time history response of the beam due to a point load $P$ (kN) moving with constant velocity $v$ (m/s) across the span. Using conventional modal analysis [23], we express the transverse displacement in terms of

the generalized coordinates $q_i(t)$ as $w(x,t) = \sum\limits_{i=1}^{\infty} \Phi_i(x) q_i(t)$ so that the equation of motion now reads as

$$EI\,\Phi_i''''(x) q_i(t) + \rho A\,\Phi_i(x)\ddot{q}_i(t) = P\,\delta(x - vt) \tag{31}$$

Next, we multiply by the eigenfrequency $\Phi_i(x)$ both sides of Equation (31) and integrate along the beam's length $[0, L]$ using the orthogonality property $\int_0^L \Phi_i(x)\Phi_j(x)\,dx = 0$. The resulting equation is

$$\ddot{q}_i(t) + \omega_i^2 q_i(t) = P\Phi_i(vt) \tag{32}$$

where the natural frequencies $\omega_i$ appear in Table 2, in the form $f_i = \omega_i\,/2\pi$.

The solution to the above equation for zero initial conditions is simply

$$q_i(t) = \frac{P}{\omega_i} \int_0^t \sin\omega_i(t - \tau)\Phi_i(v\tau)\,d\tau \tag{33}$$

which can easily be evaluated numerically by Simpson's rule. It is also possible to evaluate this convolution-type integral analytically (see Appendix A). We note that a similar approach was used by the authors [9] for the uncracked beam traversed by a moving mass, using the much simpler form for the eigenfunctions, i.e., $\Phi_i(x) = \sqrt{2/M}\sin k_i x$, where the beam's total mass $M$ is much larger than that of the moving one.

In what follows we plot in Figures 5–7 the beams transverse acceleration $\ddot{w}(x,t) = \sum_{i=1}^{\infty} \Phi_i(x)\ddot{q}_i(t)$, where the overdot $(\cdot)$ indicates a time derivative, at station $x = 11L/20$, for the following cases previously examined: (i) no crack, (ii) vertical crack length $\alpha = 10$ mm and (iii) vertical crack length $\alpha = 25$ mm. The moving load magnitude is $P = 0.5$ kN with a constant speed of $v = 0.5$ m/s, which implies that the travel time across the beam's span is 10 s. The second time derivative of the generalized functions is computed directly from the equation of motion as $\ddot{q}_i(t) = P\Phi_i(vt) - \omega_i^2 q_i(t)$. Also, the frequency plots in Figures 5–7 are in terms of the acceleration amplitude intensity, i.e., $10\cdot log_{10}\big(W(f)^2\big)$, which is a measure of energy.

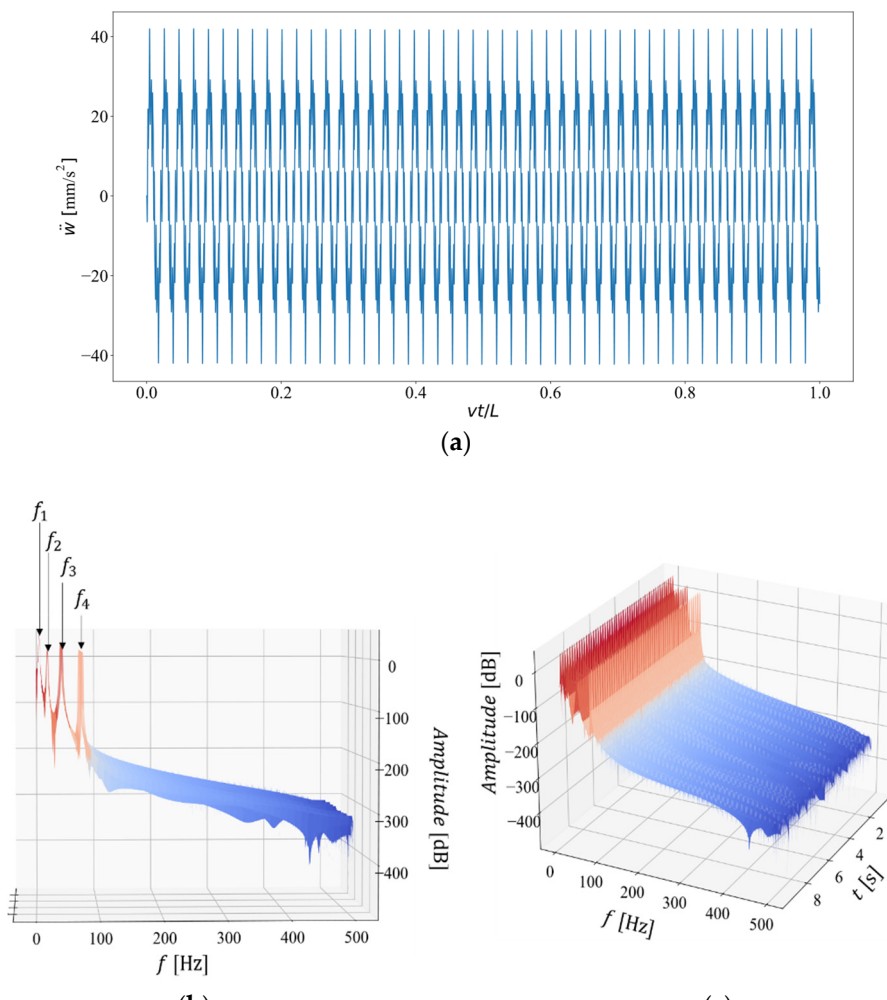

**Figure 5.** Simply-supported, uncracked beam traversed by a point force *P* with constant velocity *v*: (**a**) Acceleration time history $\ddot{w}$ (mm/s²) at station $L_1$ = 2.00 m, (**b**) Fourier transformed acceleration amplitude *W*(dB) showing the first four eigenfrequencies $fi$(Hz) and (**c**) Time-frequency spectrum of the acceleration amplitude *W*(dB). Note that colors indicate energy intensity in the signal.

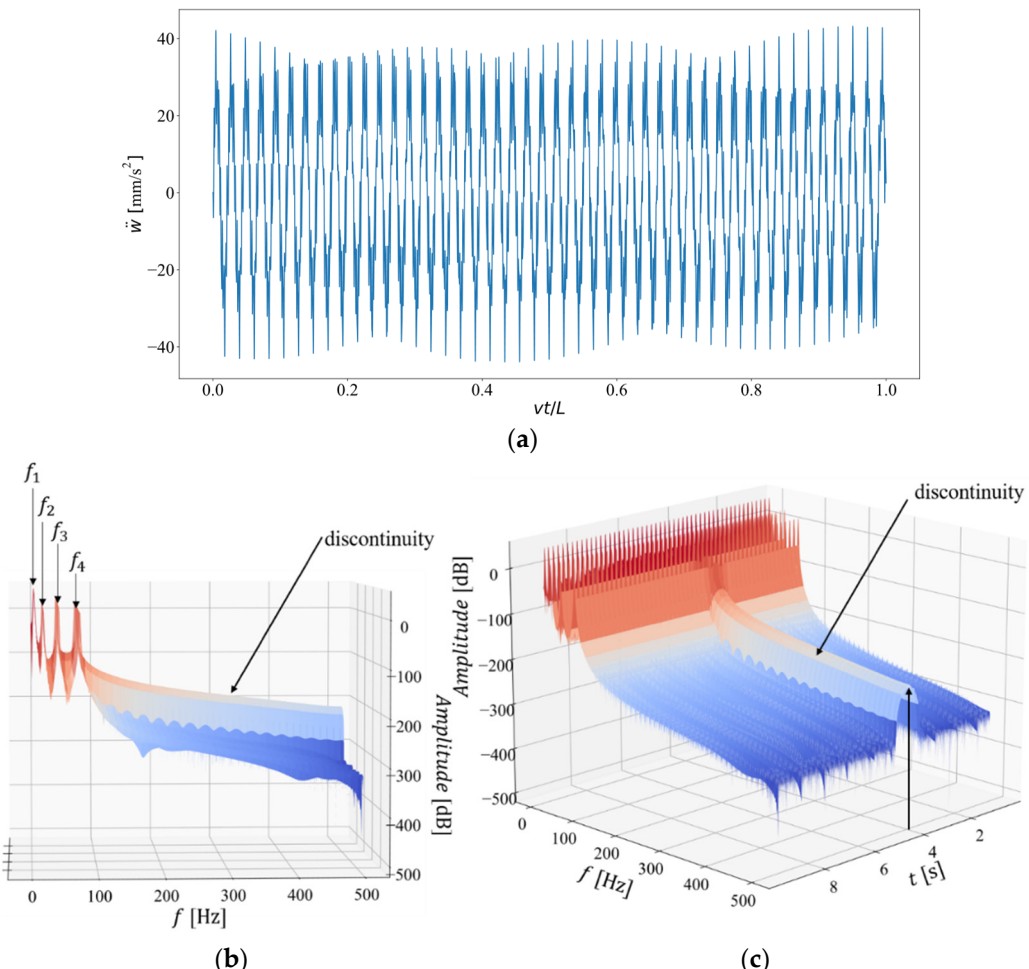

**Figure 6.** Simply-supported beam with a vertical crack of length $\alpha$ = 10 mm at station $L_1$ = 2.00 m traversed by a point force $P$ with constant velocity $v$: (**a**) Acceleration time history $\ddot{w}$(mm/s$^2$) at station $L_1$ = 2.00 m, (**b**) Fourier transformed acceleration amplitude $W$(dB) showing the first four eigenfrequencies $f_i$(Hz) and (**c**) Time-frequency spectrum of the acceleration amplitude $W$(dB) showing a discontinuity at time $t$ = 4 s. Note that colors indicate energy intensity in the signal.

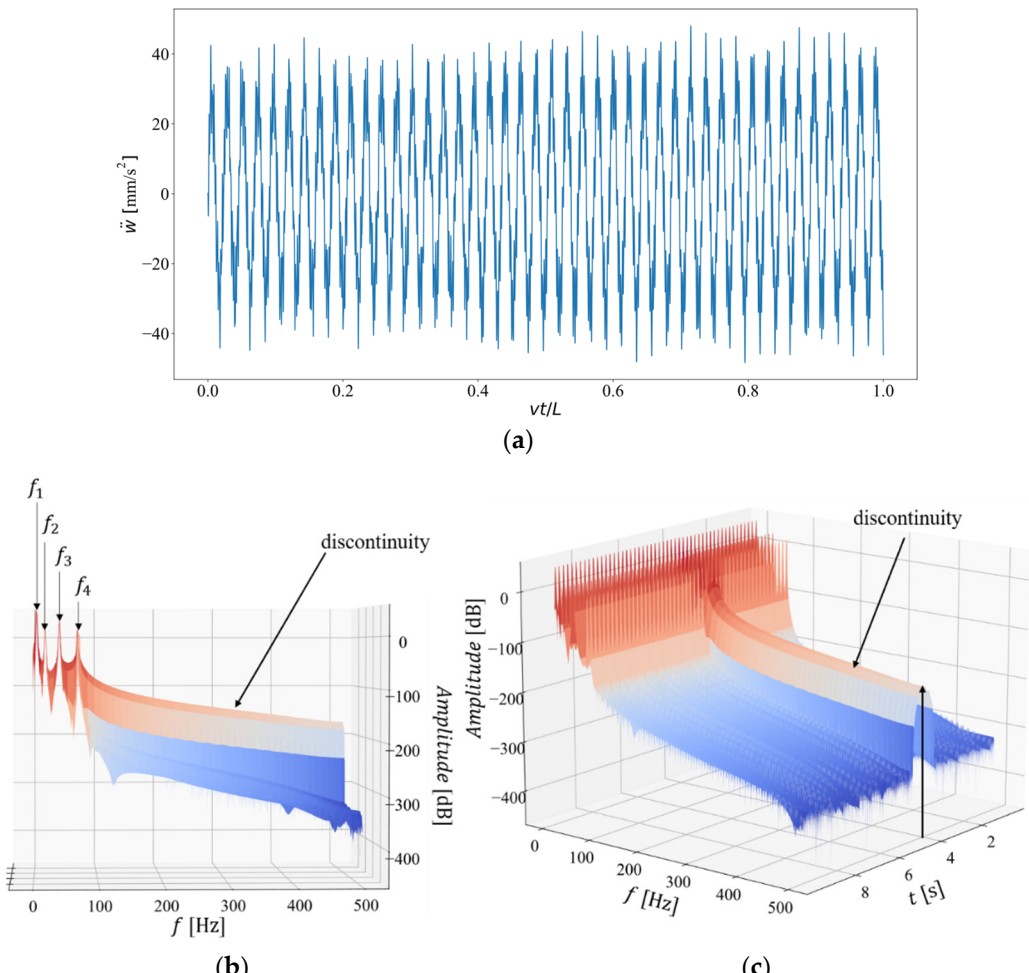

**Figure 7.** Simply-supported beam with a vertical crack of length $\alpha$ = 25 mm at station $L_1$ = 2.00 m traversed by a point force $P$ with constant velocity $v$: (**a**) Acceleration time history $\ddot{w}$(mm/s$^2$) at station $L_1$ = 2.00 m, (**b**) Fourier transformed acceleration amplitude $W$(dB) showing the first four eigenfrequencies $f_i$(Hz) and (**c**) Time-frequency spectrum of the acceleration amplitude $W$(dB) showing a discontinuity at time $t$ = 4 s. Note that colors indicate energy intensity in the signal.

## 5. Discussion and Conclusions

The important observation regarding the aforementioned figures is the appearance of a discontinuity (jump) in the Fourier-transformed accelerations due to a moving force when there is a crack in the beam's web. This is observed at time instant $t = 4$ s after the moving load enters the beam. The location of the crack can easily be determined knowing the speed of the moving load as $x_{cr} = v \cdot t = 0.5 \cdot 4 = 2$ m, which is exactly the place where the crack was inserted in the form of a spring-type discontinuity in the slope of the transverse displacement, i.e., at $x = L_1$. Furthermore, we observe in Figures 5–7 a drop in acceleration amplitude with increasing frequency, as manifested by the change of color from red to blue, which indicates lower energy intensity in the signal because the higher frequency vibrations fail to excite the beam deck as they are far removed from the dominant fundamental frequencies. Note that the discontinuity is more pronounced as the crack length grows bigger, i.e., contrast Figures 6 and 7. This behavior has important ramifications in SHM [7] as it allows the engineer to spot damage by consulting short-term Fourier transforms of recorded transient accelerations on bridge decks caused by moving loads. Note that it is hard to spot the crack by looking at transient signals form the same beam location, because the only difference between the uncracked and cracked beam accelerations is that they become less smooth in the latter case, i.e., contrast Figures 5 and 6.

In concluding, we have presented an analytical solution for the vibrations of a simply supported beam representing a bridge deck to a travelling point force, which is an idealization of a moving vehicle. From the kinematic response recorded at the deck, and particularly from its vertical accelerations at any pre-determined station, it is possible by use of the Gabor transform (also known as short-term FT) to produce time-frequency spectra that clearly show the presence of a discontinuity and allow for determining its location. This obviates the need to measure data on the travelling vehicle itself. Thus, a minimalistic data acquisition system comprising even one acceleration sensor placed at the bridge deck will yield valuable information over time to be processed for SHM purposes.

**Author Contributions:** Conceptualization, G.D.M. and G.I.D.; methodology, G.I.D. and G.D.M.; software, G.I.D.; validation, G.I.D.; formal analysis, G.D.M. and G.I.D.; resources, G.D.M.; data curation, G.I.D.; writing—original draft preparation, G.D.M. and G.I.D.; writing—review and editing, G.D.M. and G.I.D.; supervision, G.D.M.; project administration, G.D.M.; funding acquisition, G.D.M. and G.I.D. All authors have read and agreed to the published version of the manuscript.

**Funding:** The authors acknowledge the support of the German Research Foundation (DFG) through grant SM 281/20-1 and the Hellenic Foundation for Research and Innovation (HFRI) under fellowship number 6522. Any opinions, findings, conclusions or recommendations expressed in this paper are those of the authors and do not necessarily reflect the views of either the DFG or the HRFI.

**Conflicts of Interest:** The authors declare no conflict of interest.

## Glossary

List of Symbols:

1. $\rho A$ : Mass per unit length
2. $EI$: Flexure rigidity
3. $M(x)$ : Bending moment
4. $w(x, t)$: Transverse beam displacement
5. $H(x)$: Heaviside function
6. $\delta(x)$, $\delta'(x)$, $\delta''(x)$ Dirac's delta function, first and second derivatives
7. $S_i$: Functions comprising trigonometric functions
8. $\Delta$: Spatial discontinuity (jump)
9. $K_T L_i$: Equivalent spring representing the discontinuity at a node
10. $\Phi_i(x)$, $f_i$, $q_i(t)$: Eigenfunctions, eigenfrequencies, generalized coordinates

## Appendix A

Duhamel's integral appearing in Equation (33) can be evaluated analytically by expanding the trigonometric function. After some manipulation, this yields the following result:

$$q_i(t) = \frac{P}{\omega_i} \int_0^t \sin \omega_i(t - \tau) \Phi_i(v\tau) \, d\tau$$

$$= \frac{P}{\omega_i} \left( \sin \omega_i t \int_0^t \cos \omega_i \tau \, \Phi_i(v\tau) \, d\tau - \cos \omega_i t \int_0^t \sin \omega_i \tau \, \Phi_i(v\tau) \, d\tau \right)$$

If we also expand $\Phi_i(v\tau)$, the above two integrals can be divided into six simpler ones:

$$\int_0^t \cos \omega_i \tau \, c_1 \, S_1(v\tau) \, d\tau = \frac{1}{2} c_1 \left( \frac{-\omega_i \sin \omega_i t \sin k_i v\tau - k_i v \cos \omega_i t \cos k_i v\tau + k_i v}{k_i{}^3 v^2 - k_i \omega_i^2} + \frac{\omega_i \sin \omega_i t \sinh k_i v\tau + k_i v \cos \omega_i t \cosh k_i v\tau - k_i v}{k_i{}^3 v^2 + k_i \omega_i^2} \right)$$

$$\int_0^t \cos \omega_i \tau \, c_1 c_3 \, S_3(v\tau) \, d\tau = \frac{1}{2} c_1 c_3 \left( \frac{\omega_i \sin \omega_i t \sin k_i v\tau + k_i v \cos \omega_i t \cos k_i v\tau - k_i v}{k_i{}^5 v^2 - k_i{}^3 \omega_i^2} + \frac{\omega_i \sin \omega_i t \sinh k_i v\tau + k_i v \cos \omega_i t \cosh k_i v\tau - k_i v}{k_i{}^5 v^2 + k_i{}^3 \omega_i^2} \right)$$

$$\int_0^t \sin \omega_i \tau \; c_1 \; S_1(v\tau) \, d\tau = \frac{1}{2} c_1 \left( \frac{\omega_i \cos \omega_i t \sin k_i v\tau - k_i v \sin \omega_i t \cos k_i v\tau}{k_i{}^3 v^2 - k_i \omega_i^2} + \frac{-\omega_i \cos \omega_i t \; \sinh k_i v\tau + k_i v \sin \omega_i t \cosh k_i v\tau}{k_i{}^3 v^2 + k_i \omega_i^2} \right)$$

$$\int_0^t \sin \omega_i \tau \; c_1 c_3 \; S_3(v\tau) \, d\tau = \frac{1}{2} c_1 c_3 \left( \frac{-\omega_i \cos \omega_i t \sin k_i v\tau + k_i v \sin \omega_i t \cos k_i v\tau}{k_i{}^5 v^2 - k_i{}^3 \omega_i^2} + \frac{-\omega_i \cos \omega_i t \sinh k_i v\tau + k_i v \sin \omega_i t \cosh k_i v\tau}{k_i{}^5 v^2 + k_i{}^3 \omega_i^2} \right)$$

$$\int_{\frac{L_1}{v}}^t \cos \omega_i \tau \; c_1 c_2 \; S_1 \quad (v\tau - L_1) \, d\tau$$

$$= \frac{1}{2k_i} c_1 c_2 \left( \frac{1}{k_i{}^2 v^2 - \omega_i^2} \left( -k_i v \cos k_i (L_1 - vt) \cos \omega_i t + k_i v \cos \frac{L_1 \omega_i}{v} \right. \right.$$
$$+ \omega_i \sin k_i (L_1 - vt) \sin \omega_i t) \quad + \frac{1}{k_i{}^2 v^2 + \omega_i^2} \left( k_i v \cosh k_i (L_1 - vt) \cos \omega_i t - k_i v \cos \frac{L_1 \omega_i}{v} \right.$$
$$\left. \left. + \omega_i \sinh k_i (L_1 - vt) \sin \omega_i t) \right) \right.$$

$$\int_{\frac{L_1}{v}}^t \sin \omega_i \tau \; c_1 c_2 \; S_1 \quad (v\tau - L_1) \, d\tau$$

$$= \frac{1}{2k_i} c_1 c_2 \left( \frac{1}{k_i{}^2 v^2 - \omega_i^2} \left( \omega_i \cos \; k_i L_1 \cos \omega_i t \sin k_i vt \right. \right.$$
$$- \cos k_i vt \left( \omega_i \sin \; k_i L_1 \cos \omega_i t + k_i v \cos \; k_i L_1 \sin \omega_i t \right) + k_i v \left( -\sin k_i L_1 \sin k_i vt \sin \omega_i t + \sin \frac{L_1 \omega_i}{v} \right) \right)$$
$$\left. + \frac{1}{k_i{}^2 v^2 + \omega_i^2} \left( k_i v \cosh k_i (L_1 - vt) \sin \omega_i t - k_i v \sin \frac{L_1 \omega_i}{v} + \omega_i \sinh k_i (L_1 - vt) \cos \omega_i t \right) \right)$$

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
