# Peer review of "On the Detection of Fracture within Vibrating Beams Traversed by a Moving Force"

_infrastructures, doi:10.3390/infrastructures7070093_

Round 1
Reviewer 1 Report
This paper explored the influence of existing cracks in the beam on the bridge continuity under dynamic loadings. The research content could provide a significant reference for guiding bridge detection. However, before it was accepted for publication, some necessary modifications still needed to be carried out. They are:
(1) Although the abstract fully introduced this study's research purpose and methods, the content of their main conclusions is insufficient. Please supplement. The keywords could not cover the full-text content well. Can make some improvements.
(2) The introduction covered the background knowledge and theoretical basis of the research basically. However, some research on the influence of cracks on the performance of structural components needed to be discussed to enrich the content. In addition, please describe the research work done in detail and clarify the steps and procedures.
(3) Please give a unified description of the specific symbols appearing in the formula.
(4) Line 228: how to obtain the necessary boundary conditions?
(5) The quality of the figure was relatively high, but some language expressions in these figures needed to be optimized. For example, what were the specific meanings of different colors in Figures 5, 6, and 7?
(6) The discussion and conclusion were too short. It was suggested to expand them in combination with the existing research results.
(7) The number of references was insufficient, and some were long-standing. Please add the relevant literature published on MDPI in recent years.
Author Response
Manuscript ID: Infrastructures-1792720
Type of manuscript: Article
Title: On the Detection of Fracture within Vibrating Beams Traversed by a Moving Force
Authors: Georgios I. Dadoulis, George D. Manolis
REPLY TO THE REVIEW
We wish to thank the reviewers for their effort in reviewing the above manuscript and for providing constructive criticism and recommendations. The manuscript has now been modified accordingly, with all modifications highlighted in colour in the revised version. Furthermore, the writing at places has been improved. In what follows, we respond to the comments raised by each reviewer.
Reviewer 1
- The Abstract became more focused in its conclusions and the Keywords list was enriched
- The Introduction was enriched in terms of additional, contemporary references addressing practical engineering concerns on the effect of cracks on the structural performance of beam decks. The flowchart of the work done in this paper was also improved.
- A List of Symbols was added.
- The boundary conditions in line 228 were clarified.
- The captions in Figs. 5-7 were clarified.
- Both Discussion and Conclusions were expanded.
- This was addressed in conjunction with item (2).
Reviewer 2 Report
The authors presented a study on detection of fracture within vibrating beams traversed by a moving force. The manuscript is finely written and the concept is clear. However the reviewer has the following comments that must be addressed before its acceptance.
(1) The idea to use flexible or rotational spring for cracks is not new and has been used in crack detections in literatures. The results presented by the authors follow the same approach and do not have new findings in the reviewer's point of view.
(2) The authors summarized the classical Euler-Bernouli beam mechanics models of fracture well.
(3) The presented FFT amplitude versus time and frequency is the same as the time-frequency spectrum widely used in SHM and damage detection. The reviewer does not see the new findings in this aspect. The only interesting point in the manuscript might be to obtain the frequencies and mode shapes of the cracked beam through the Laplace transformation of the classical Euler-Bernouli model of cracked beams.
(4) Many literatures on time-frequency analysis of crack detection and the FFT analysis of cracked beams under traveling loads or trucks should be included and cited, such as "Possibility of bridge inspection through drive-by vehicles", Applied Science, 2020, 11(1); "Dynamic responses of prestressed bridge and vehicle through bridge-vehicle interaction analysis", Engineering structures, 2015, 87, 116-125.
Author Response
Manuscript ID: Infrastructures-1792720
Type of manuscript: Article
Title: On the Detection of Fracture within Vibrating Beams Traversed by a Moving Force
Authors: Georgios I. Dadoulis, George D. Manolis
REPLY TO THE REVIEW
We wish to thank the reviewers for their effort in reviewing the above manuscript and for providing constructive criticism and recommendations. The manuscript has now been modified accordingly, with all modifications highlighted in colour in the revised version. Furthermore, the writing at places has been improved. In what follows, we respond to the comments raised by each reviewer.
Reviewer 2
- True statement. We have relied on this engineering-type approach of modelling cracks in bridge decks as discontinuities between two consecutive segments connected by springs. Our contribution is in manipulating the analytical solution we derived to the point where it is possible to detect an anomaly in the displacement spectra of the beam as it is being traversed by a moving load, which indicates the presence of a discontinuity, and from the time this occurs, to also pin-point its location.
- Thank you for this comment. It should be noted that moving loads on beams is an extensive subject dating from form the 1960’s (L. Fryba’s book on Vibrations of Solids and Structures under Moving Loads translated in 1972 is a classical reference). Also, the book by Bayer and Dyniewicz (2012) is a superb source of information on the numerical treatment of the dynamic response of beams to moving loads. These have been added, along with other relevant references as suggested by the reviewer.
- There are a couple of methods for solving this problem which actually has two discontinuities, one spatial (the crack) and one temporal (the moving point load itself). We note in passing that if the mass of the moving load is sizeable, it must be taken into account as it will change the dynamic properties of the beam. These two discontinuities coalesce at the instant the load goes over the crack and this can be viewed as a singular point. We mention alternative methods for handling this problem, but the eigenvalue problem solved with the aid of the Laplace transform turned out to be the best choice. It is true that time-frequency spectra are often used, not only in SHM but in earthquake engineering as well, but in our case they were employed for crack detection.
- We thank the reviewer for pointing out some interesting references, which have been included in the revised manuscript.
Round 2
Reviewer 1 Report
Ok, it can be accepted for publication